# NADPH oxidase-mediated redox signaling promotes oxidative stress resistance and longevity through *memo-1* in *C. elegans*

Collin Yvès Ewald[1,2,3,4,5][\*][†], John M Hourihan[3,4,5][†], Monet S Bland[3,4,5], Carolin Obieglo[3,4,5], Iskra Katic[2], Lorenza E Moronetti Mazzeo[3,4,5], Joy Alcedo[2,6], T Keith Blackwell[3,4,5][\*], Nancy E Hynes[2][\*]

[1]Department of Health Sciences and Technology, Eidgenössische Technische Hochschule (ETH) Zürich, Zürich, Switzerland; [2]Friedrich Miescher Institute for Biomedical Research, University of Basel, Basel, Switzerland; [3]Department of Genetics, Harvard Medical School, Boston, United States; [4]Joslin Diabetes Center, Boston, United States; [5]Harvard Stem Cell Institute, Cambridge, United States; [6]Department of Biological Sciences, Wayne State University, Detroit, United States

**Abstract** Transient increases in mitochondrially-derived reactive oxygen species (ROS) activate an adaptive stress response to promote longevity. Nicotinamide adenine dinucleotide phosphate (NADPH) oxidases produce ROS locally in response to various stimuli, and thereby regulate many cellular processes, but their role in aging remains unexplored. Here, we identified the *C. elegans* orthologue of mammalian mediator of ErbB2-driven cell motility, MEMO-1, as a protein that inhibits BLI-3/NADPH oxidase. MEMO-1 is complexed with RHO-1/RhoA/GTPase and loss of *memo-1* results in an enhanced interaction of RHO-1 with BLI-3/NADPH oxidase, thereby stimulating ROS production that signal via p38 MAP kinase to the transcription factor SKN-1/NRF1,2,3 to promote stress resistance and longevity. Either loss of *memo-1* or increasing BLI-3/NADPH oxidase activity by overexpression is sufficient to increase lifespan. Together, these findings demonstrate that NADPH oxidase-induced redox signaling initiates a transcriptional response that protects the cell and organism, and can promote both stress resistance and longevity.

**\*For correspondence:** collin-ewald@ethz.ch (CYE); keith.blackwell@joslin.harvard.edu (TKB); nancy.hynes@fmi.ch (NEH)

[†]These authors contributed equally to this work

## Introduction

How reactive oxygen species (ROS) affect aging is a fundamental issue in biology (*Back et al., 2012a*; *Krause, 2007*; *Hekimi et al., 2011*; *Ristow, 2014*; *Ristow and Schmeisser, 2014*; *Kawagishi and Finkel, 2014*; *Riera and Dillin, 2015*; *Durieux et al., 2011*; *Balaban et al., 2005*; *Melov, 2002*; *Shadel, 2014*; *Sena and Chandel, 2012*). Chronic exposure to ROS leads to cellular damage and age-associated diseases, including Alzheimer's disease, Parkinson's disease, cancer, diabetes, cardiovascular diseases, and chronic inflammation. By contrast, low or acute ROS exposure mobilizes protective mechanisms and increases lifespan in *S. cerevisiae* (*Pan et al., 2011*; *Mesquita et al., 2010*; *Schroeder et al., 2013*), *D. melanogaster* (*Albrecht et al., 2011*), *C. elegans* (*Schulz et al., 2007*; *Doonan et al., 2008*; *Yang and Hekimi, 2010*; *Schmeisser et al., 2013*; *Lee et al., 2010*), and rodents (*Lapointe and Hekimi, 2008*; *Liu et al., 2005*), and has been associated with health benefits in humans (*Ristow et al., 2009*, *2014*). This phenomenon is conceptualized as mitochondrial hormesis or mitohormesis (*Ristow and Schmeisser, 2014*).

Hormesis is defined as the induction of protective mechanisms under exposure to low doses of stressful agents, which at higher or prolonged exposures are harmful. Ninety percent of cellular ROS

arise as a by-product of mitochondrial oxidative phosphorylation (*Halliwell and Gutteridge, 2007*). Moderate mitochondrial dysfunction leads to elevation of mitochondrial-derived ROS, activating protective mechanisms (mitohormesis) and promoting longevity (*Ristow and Schmeisser, 2014*). Several longevity interventions, such as dietary restriction or reduced insulin/IGF-1 signaling, are associated with an increase in mitochondrial ROS levels, which acts as a retrograde signal to increase lifespan (*Schulz et al., 2007*; *Weimer et al., 2014*; *Zarse et al., 2012*).

One mechanism by which ROS affect cellular signaling is by specifically and reversibly reducing/oxidizing reactive thiol-groups on cysteine residues, thereby modifying protein functions, which is also known as redox signaling. For instance, ROS promote receptor tyrosine kinase (RTK) signaling by oxidizing a cysteine residue in protein-tyrosine phosphatase 1 (PTP1), thereby transiently inactivating its phosphatase activity (*Salmeen et al., 2003*; *Goldstein et al., 2005*) and potentiating the activity of its partner RTK. Localized ROS that act as signals do so at short range (~5–20 μm; [*Winterbourn, 2008*]) with a half-life of ~1 ms (*D'Autréaux and Toledano, 2007*), in part due to high intracellular concentration of the antioxidant glutathione (1–10 mM; [*Meister, 1988*]), which keeps the cytosol in a reduced environment (*Gilbert, 1990*; *Romero-Aristizabal et al., 2014*; *Lambeth and Neish, 2014*). Hence, pools of localized ROS must be rapidly generated for redox signaling to occur.

In addition to ROS derived as a by-product of mitochondrial oxidative phosphorylation, cells have membrane-associated enzymes that generate ROS, using nicotinamide adenine dinucleotide phosphate (NADPH) as an electron donor to produce a local ROS micro-environment (*Bedard and Krause, 2007*). In general, NADPH oxidases form complexes with subunits required for their stability and activation (*Bedard and Krause, 2007*). For instance, upon stimulation of cell surface receptors, guanosine-trisphospate (GTP) bound Rho-guanosine-triphosphatase (GTPase) family members and p21-activated kinase-1 (PAK1) must be recruited to the NADPH oxidase complex to generate ROS (*Hurd et al., 2012*).

Mammals have seven NADPH oxidase family members, which have been found in almost every tissue and are localized at cellular membranes and within intracellular compartments, such as endosomes and endoplasmic reticulum (ER) (*Bedard and Krause, 2007*; *Krause, 2007*). Mammalian NADPH oxidases have been implicated in a wide range of normal physiological functions, (*Bedard and Krause, 2007*; *Krause, 2007*), as well as in diseases that include cancer (*Truong and Carroll, 2012*).

NADPH oxidase-generated ROS have been shown to act as a second messenger to regulate migration of metastasis-committed-cancer cells and as a chemoattractant for immune cells during wound healing (*Stanley et al., 2014*; *Hurd et al., 2012*). Mediator of ErbB2 driven cell motility (Memo1) has been shown to play an important role in migration of breast cancer cells and is needed for robust metastatic dissemination from primary tumors to lungs (*Marone et al., 2004*; *MacDonald et al., 2014*). During the migratory process Memo1 interacts with Rho GTPase to dynamically reorganize actin and microtubule fibers (*Zaoui et al., 2008*), and has also been linked to NADPH oxidase activity in breast cancer cells (*MacDonald et al., 2014*). However, whether NADPH oxidase generated ROS have a biological function during aging is unknown. Here, we used the model organism *Caenorhabditis elegans* to investigate the role of NADPH oxidase generated ROS in aging. The nematode *C. elegans* provides the advantages of genetic tractability, and of being transparent that allows *in vivo* non-invasive visualization of transgenic fluorescent probes that measure ROS levels (*Back et al., 2012b*). Moreover, *C. elegans* has a short lifespan, making it ideal to gain mechanistic insights into the aging process. We found that loss of the *C. elegans memo-1/C37C3.8* leads to elevated ROS levels generated by BLI-3/NADPH oxidase, which activates an adaptive detoxification system regulated by the transcription factor SKN-1/Nrf1,2,3 in promoting organismal-wide oxidative stress resistance and longevity.

## Results

### Loss of *memo-1* increases lifespan and oxidative stress resistance

The nematode *C. elegans* encodes a gene (C37C3.8) that shares 153 out of 297 amino acids (52%) identity with human Memo1 (*Figure 1—figure supplement 1A*), which we named *memo-1* (Mediator of ErbB2-driven cell motility-like protein; wormbase.org). The *C. elegans memo-1* gene product

is expressed in the embryo and larval stages and through adulthood in many neuronal and non-neuronal cells in the head and tail, spermatheca, distal tip cells, anchor cell, and intestine (*jxEx8* [P*memo-1*::GFP] and *ldEx112* [MEMOfosmid::GFP]; *Figure 1A* and *Figure 1—figure supplement 1B–I*). To gain insight into the biological function of *memo-1* in *C. elegans*, we took a reverse genetics approach using *memo-1* RNA interference (*memo-1(RNAi)*) or *memo-1(gk345)* putative null mutants (*memo-1(-)*; *Figure 1B*) and measured longevity. Both *memo-1(RNAi)* and *memo-1(gk345)* mutants showed a 7–38% increase in lifespan compared to wild type (*Figure 1C*, *Supplementary file 1*). We also tested these mutants for several pathways related to Memo1 activities in vertebrates (*Sorokin and Chen, 2013*; *Marone et al., 2004*), which include phenotypes linked to the epidermal growth factor receptor (EGFR) and insulin/IGF-1 receptors (*Figure 1—figure supplement 2E*), and to cell migration (*Figure 1—figure supplement 2F–J*). We found that *memo-1(-)* mutants behaved like the wild-type strain. To exclude any developmental effects of *memo-1* loss, we treated wild-type animals with *memo-1(RNAi)* starting from the first day of adulthood and showed that this was sufficient to increase lifespan (*Figure 1D* and *Supplementary file 1*).

Dietary restriction extends lifespan across essentially all eukaryotes (*Kenyon, 2010*). Since MEMO-1 is expressed around the pharynx (*Figure 1—figure supplement 1C*), an organ that regulates food intake, we measured pharyngeal pumping rates of *memo-1*(-) mutants, finding them to be similar to those of wild type (*Figure 1—figure supplement 3A*). Reducing germline stem cell number also increases *C. elegans'* lifespan (*Hsin and Kenyon, 1999*) and MEMO-1 is expressed in cells associated with the germline (*Figure 1—figure supplement 1F and H*). However, *memo-1(-)* mutants did not show germ-line defects (*Figure 1—figure supplement 2F and G*). Not only did they have a normal brood size (*Figure 1—figure supplement 3B*), they also showed no upregulation of genes implicated in reduced germline stem cell-mediated longevity (*Figure 1—figure supplement 3C*).

To identify cellular processes that promote the *memo-1(-)*-longevity phenotype, we performed quantitative RT-PCR on selected target genes involved in known longevity-promoting processes, comparing *memo-1(-)* mutants with wild type (*Figure 1—figure supplement 3C–G*). We concentrated on stress response pathways, since longevity frequently correlates with increased stress resistance (*Shore and Ruvkun, 2013*). Several genes involved in the oxidative stress response were strongly upregulated (3–10 fold) in the *memo-1* mutants or by *memo-1(RNAi)*, compared to wild type or empty vector control strains (*Figure 1E–I*, *Figure 1—figure supplement 3G*). By contrast, genes involved in the heat shock response, the unfolded protein response (UPR) and other longevity-promoting processes were not altered in *memo-1(-)* animals (*Figure 1—figure supplement 3C and E*). Importantly, downregulation or loss of *memo-1* by RNAi or mutation increased oxidative stress resistance, correlating with the observed changes in gene expression (*Figure 1J*, *Supplementary file 2*). By contrast, its loss did not increase resistance to heat shock (*Figure 1—figure supplement 3H*). Taken together, these results suggest that a elevated oxidative stress resistance might be an important factor in *memo-1(-)*-longevity.

## Loss of *memo-1* activates p38 MAPK signaling and the oxidative stress response transcription factor SKN-1

The major oxidative stress response is orchestrated by the transcription factor SKN-1, a homologue of the NRF1/2/3 (Nuclear factor-erythroid-related factor) bZIP transcription factor family (*An and Blackwell, 2003*). Several of the oxidative stress response genes that are upregulated by *memo-1* knockdown or mutation are direct transcriptional targets of SKN-1 (*Figure 2—source data 1*). Thus, we examined the role of *skn-1* in the *memo-1* phenotype. Knockdown of *skn-1* by RNAi abolished the longevity phenotype of *memo-1(-)* mutants (*Figure 2A*; *Supplementary file 1*). Conversely, *memo-1(RNAi)* knockdown increased lifespan of wild-type animals but not of *skn-1(-)* mutants (*Figure 2B*; *Supplementary file 1*). These results suggest that the extended lifespan of animals that lack *memo-1* is mediated through SKN-1.

To determine whether loss of *memo-1* activates SKN-1, we examined the cellular localization of GFP-tagged SKN-1. Under normal conditions SKN-1 is largely retained in the cytoplasm; following oxidative stress, SKN-1 accumulates in the nucleus to initiate transcription of target genes (*An and Blackwell, 2003*). RNAi-mediated *memo-1* knockdown caused a significant increase in nuclear SKN-1::GFP (*Figure 2C and D*). Under oxidative stress conditions, SKN-1 phosphorylation/activation is mediated by p38 mitogen-activated protein kinase (MAPK; [*Inoue et al., 2005*]), which is phosphorylated and

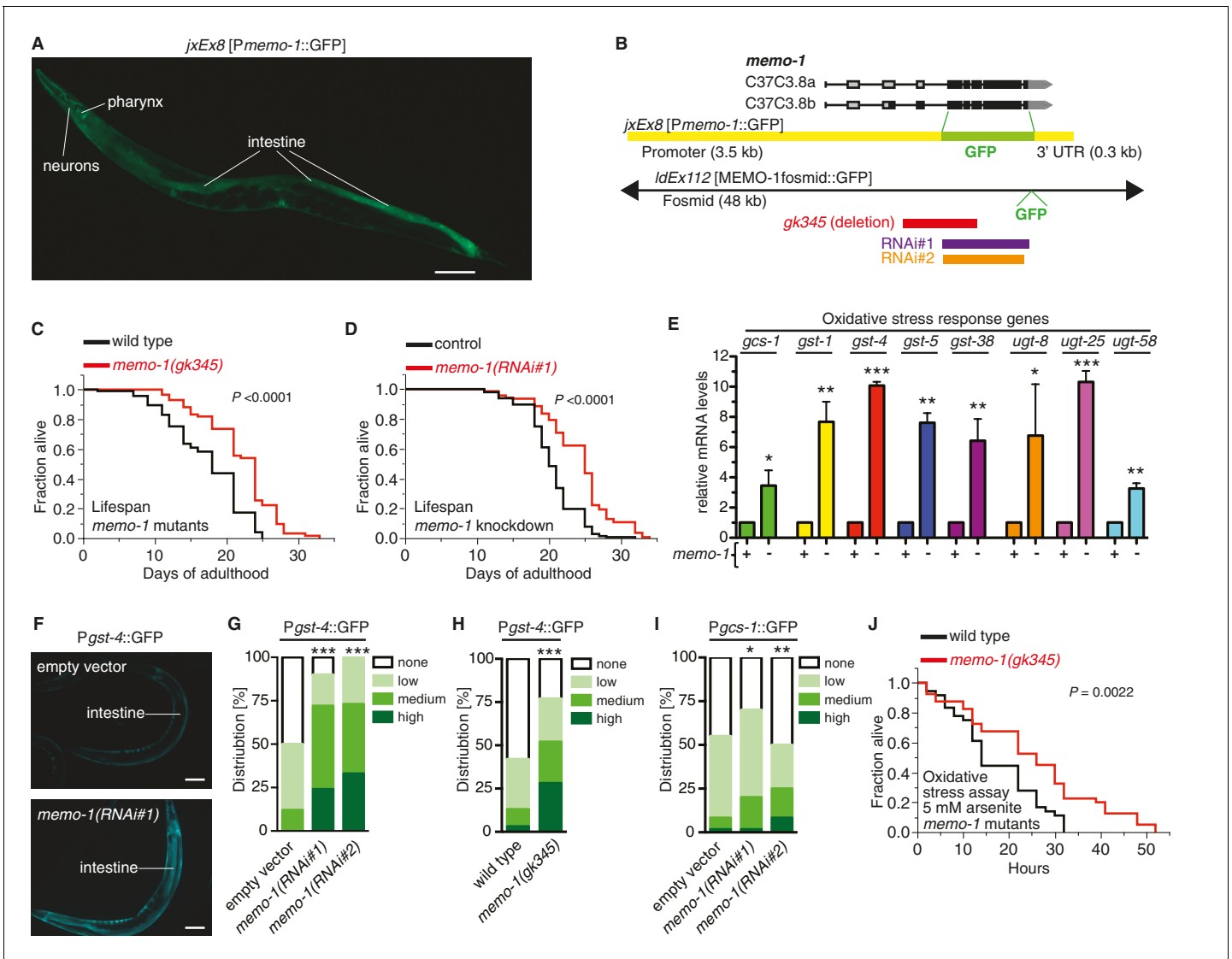

**Figure 1.** Loss of *memo-1* leads to increased ROS. (A) Transcriptional fusion of the *memo-1* promoter (P*memo-1*) with GFP (*jxEx8* [P*memo-1*::GFP]) shows that *memo-1* is expressed in neurons, pharyngeal cells, and intestine in adult *C. elegans*. Anterior to the left, ventral side down. Scale bar = 100 µm. (B) Genomic organization of the *C37C3.8 (memo-1)* locus (gray is untranslated UTR; black are translated exons; adapted from wormbase.org). The *memo-1* gene encodes two isoforms (C37C3.8a and C37C3.8b), whereby C37C3.8b is predicted to be 48 amino acids longer than C37C3.8a (297 amino acids). The *gk345* allele (red) is a 915 bp deletion. RNAi#1 clone (purple) and RNAi#2 clone (orange) are from Vidal- and Ahringer RNAi libraries, respectively. See Materials and methods for more details. (C) *memo-1(gk345)* mutants show a 27% increase in mean lifespan compared to wild type (N2) at 20°C. *P* value determined by log-rank. Statistics and additional lifespan data are in *Supplementary file 1*. (D) Knockdown of *memo-1(RNAi#1)* starting on the first day of adulthood in RNAi-sensitive animals (*rrf-3(pk1426)*) increases mean lifespan by 20% compared to empty RNAi vector control (L4440) at 20°C. *P* value determined by log-rank. Statistics and additional lifespan data are in *Supplementary file 1*. (E) *memo-1(RNAi#2)* treated wild type (N2) (-) have higher mRNA expression levels of the oxidative stress response genes, such as glutamine cysteine synthetase (*gcs-1*), glutathione-S-transferase (*gst-1, 4, 5, 38*), and uridine 5'-diphospho-glucoronosyltransferase (*ugt-8, 25, 58*), compared to empty vector treated wild type (N2) (+), determined by qRT-PCR. 3 replicates of >1000 mixed staged worms per condition were analysed. Data are represented as mean ± s.e.m. *P* value * <0.05, ** <0.001, *** <0.0001 relative to wild type or control, by one sample *t*-test, two-tailed, hypothetical mean of 1. (F–I) Loss of *memo-1* increases the expression of oxidative stress response genes *gst-4* and *gcs-1* in the intestine. (F) shows representative pictures of *dvIs19* [P*gst-4*::GFP] transgenic adult *C. elegans* treated with empty vector (upper picture; category: none) or *memo-1(RNAi#1)* (lower picture; category: high). Anterior to the right, ventral side up. Scale bar = 100 µm. (G–I) Quantification of transgenic worms containing the promoter of *gst-4* or *gcs-1* fused with GFP (*dvIs19* [P*gst-4*::GFP] and *ldIs003* [P*gcs-1*::GFP]. Scoring is described in Material and methods. Three trials are shown with N > 60 for each condition and trial. *P* value by chi[2] (* <0.05; ** <0.001; ***p<0.0001). (J) Survival of one-day old adult *memo-1(gk345)* mutants or wild type (N2) in sodium arsenite (5 mM) was assayed. *P* value determined by log-rank. Statistics and additional oxidative stress data either with arsenite or *tert*-butyl hydrogen peroxide are shown in *Supplementary file 2*.

*Figure 1 continued on next page*

*Figure 1 continued*

The following figure supplements are available for figure 1:

**Figure supplement 1.** MEMO-1 is a conserved protein that is expressed in many tissues in *C. elegans*.

**Figure supplement 2.** Reverse genetics approach to determine *memo-1* function.

**Figure supplement 3.** Reduced *memo-1* function induces oxidative stress gene expression.

activated by SEK-1/MAPKK (*Inoue et al., 2005*; *Hourihan et al., 2016*). Importantly, reducing *memo-1* function either by RNAi or mutation resulted in higher levels of phospho-p38 MAPK (*Figure 2E and F*), and loss of *sek-1* completely abolished *memo-1(-)* mediated-longevity (*Figure 2G*). Since *sek-1* and *skn-1* are important for the defense against pathogenic bacteria (*Hoeven et al., 2011*), and bacterial proliferation in the *C. elegans'* intestine contributes to death of the animal (*Garigan et al., 2002*), we investigated the lifespan of *memo-1* mutants on heat-killed bacteria. We found that *memo-1* mutants remain long-lived compared to wild type (*Supplementary file 1*). Taken together, these results show that *memo-1* loss stimulates the p38 MAPK oxidative stress response pathway (*Figure 2H*), which is required for the longevity phenotype.

## Loss of *memo-1* leads to increased ROS

The increase in the oxidative stress response pathway observed in the preceding experiments suggested that ROS levels might be altered in *memo-1* animals. To examine this, we measured intracellular ROS in living worms using a diffusible fluorescent probe (chloromethyl derivative CM-$H_2$DCFDA) and observed a 2-fold increase upon treatment of wild-type *C. elegans* with *memo-1 (RNAi)* (*Figure 3A*). Moreover, compared to control worms, transgenic worms that express the HyPer sensor (*jrIs1* [*Back et al., 2012b*]) and have been treated with *memo-1(RNAi)* have a 5-fold increase in endogenous hydrogen peroxide levels (*Figure 3B*), while *memo-1* mutant or RNAi-treated animals have a 2-fold increase in hydrogen peroxide, according to the fluorescent probe AmplexRed (*Figure 3C and D*). Interestingly, ROS levels in the *memo-1(RNAi)* and the *memo-1 (RNAi); sek-1(RNAi)* animals were elevated to a similar level (*Figure 3E* and *Figure 3—figure supplement 1A and B*), suggesting that the SKN-1 adaptive response did not necessarily reduce the overall ROS load. Additionally, *memo-1(-)*-impaired animals that lack ROS protection in a *skn-1(-)* (*Figure 2B*) or *sek-1(-)* mutant background (*Figure 2G*) did not live shorter than either *skn-1(-)* or *sek-1(-)* single mutants in the absence (*Figure 2B and G*) or presence of the oxidative stress-inducing agent 5 mM arsenite (*Supplementary file 1* and *2*). Taken together, these results suggest that the increased ROS levels in animals that lack *memo-1* are not necessarily detrimental.

An intriguing possibility suggested by the data is that *memo-1(-)*-induced ROS might have longevity-promoting activities analogously to mitochondrially-derived ROS. To investigate this hypothesis, we neutralized *memo-1(RNAi)*-produced ROS by treatment with the antioxidant glutathione (GSH) (*Figure 3G* and *Figure 3—figure supplement 1C*). GSH treatment during adulthood had no effect on wild-type lifespan, but abolished *memo-1(-)*-induced longevity (*Figure 3H*, *Supplementary file 1*), suggesting an essential role for *memo-1(-)*-induced ROS in this process. Furthermore, GSH treatment abolished the induction of the SKN-1 target gene *gst-4* (*Figure 3I*), indicating that the *memo-1(RNAi)*-induced ROS are responsible for SKN-1 activation.

## Excess ROS in *memo-1(-)* animals is generated by the dual oxidase BLI-3

SKN-1 has been implicated in ROS-induced hormesis and ROS signaling (*Blackwell et al., 2015*). Indeed, mitochondrially-generated ROS have been observed to increase lifespan via SKN-1 (*Schmeisser et al., 2013a*; *Weimer et al., 2014*; *Schmeisser et al., 2013b*; *Hunt et al., 2011*). To determine if excess ROS in *memo-1(-)* mutants are generated from mitochondria, we used a specific mitochondrial ROS-detecting reagent, MitoTracker Red CM-$H_2$X (*Weimer et al., 2014*), revealing that the *memo-1(RNAi)* animals had similar mitochondrial ROS levels as the controls (*Figure 4A*). Furthermore, a mitochondria specific-antioxidant, MitoTEMPO, neither suppressed the higher ROS levels nor elevated the *gst-4* expression levels (*Figure 4C*) seen in *memo-1(RNAi)*-treated animals

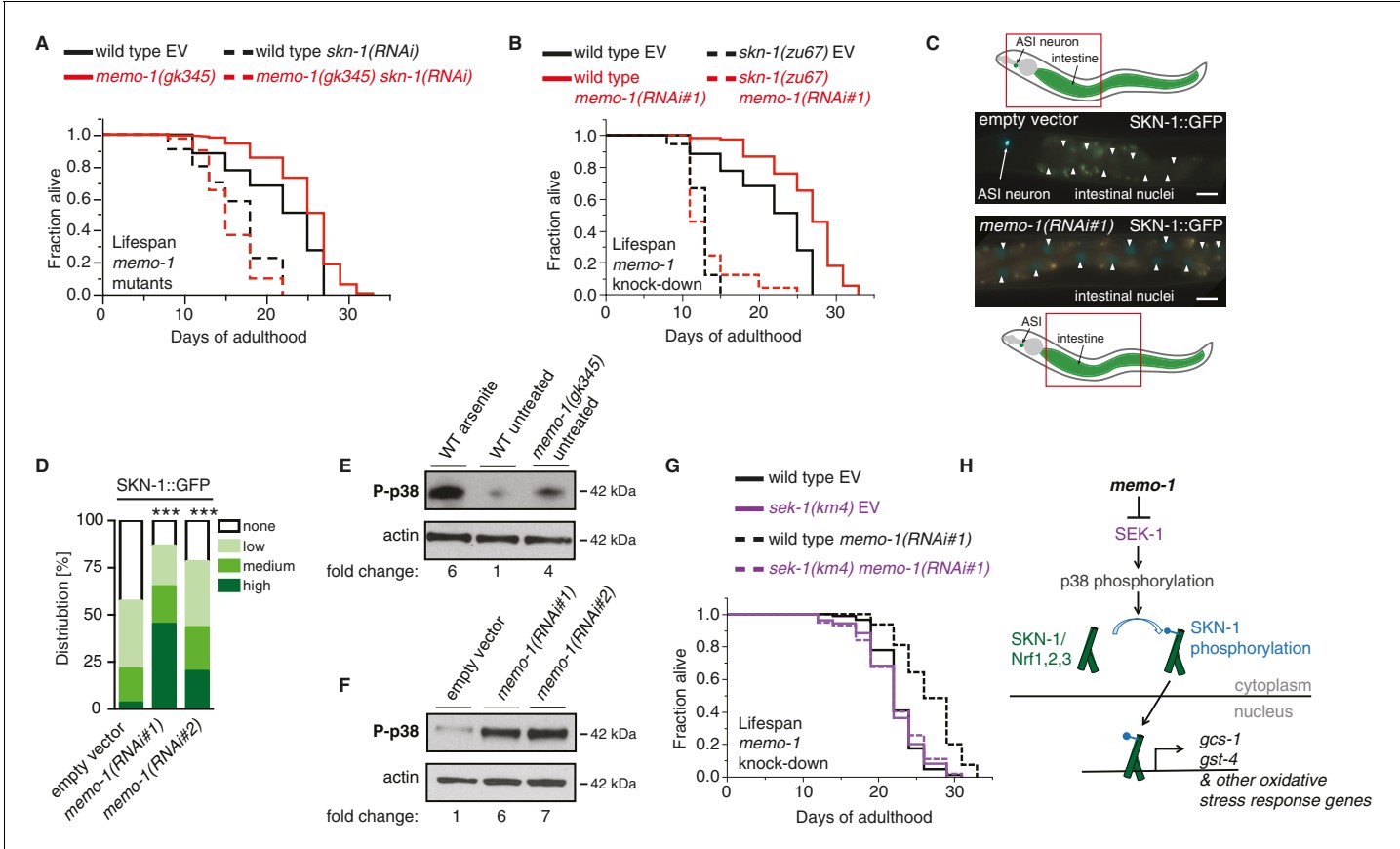

**Figure 2.** The oxidative stress response pathway is activated and is required for *memo-1(-)* mediated longevity and oxidative stress resistance. (**A**) Knockdown of *skn-1* starting on the first day of adulthood abolishes the increased lifespan of *memo-1(gk345)* mutants. For statistical details and additional lifespans see **Supplementary file 1**. (**B**) Knockdown of *memo-1(RNAi#1)* starting on the first day of adulthood increases lifespan of wild type (N2) animals, but failed to increase lifespan of *skn-1(zu67)* loss-of-function mutants. For statistical details and additional strains see **Supplementary file 1**. (**C**) Schematics and representative pictures of transgenic L4 animals expressing a translational fusion of SKN-1 protein tagged with GFP (*ldIs007* [SKN-1::GFP]), treated either with empty vector (upper picture; no SKN-1::GFP in intestine = score none) or *memo-1(RNAi#1)* (bottom picture; SKN-1:: GFP in all intestinal nuclei = score high). Scale bar = 20 µm. Triangles indicate intestinal nuclei. (**D**) Quantification of SKN-1::GFP in L4 transgenic animals: Knockdown of *memo-1(RNAi #1 or #2)* starting from the egg stage induced translocation of SKN-1::GFP into the intestinal nuclei. N > 60, three merged trials. *** <0.0001 *P* values were determined by Chi[2] test. Scoring is described in Material and Methods. (**E**) p38 mitogen-activated protein kinase is phosphorylated (P-p38 MAPK) upon oxidative stress (WT arsenite; 5 mM sodium arsenite for 10 min) or in untreated *memo-1(gk345)* mutants, compared to untreated wild type (WT untreated). (**F**) Knockdown of *memo-1(RNAi #1 or #2)* for two generations led to an increase p38 phosphorylation levels compared to empty RNAi vector control wild type animals. For (**E–F**) Total extracts from >1000 L4 animals in each condition were analyzed by western for levels of P-p38. Actin was used as a loading control. Fold change indicates the relative P-p38 MAPK to actin levels compared to control (E: WT untreated; F: WT empty RNAi vector control). (**G**) Knockdown of *memo-1(RNAi#1)* starting on the first day of adulthood increases lifespan of wild type (N2) animals, but failed to increase lifespan of *sek(km4)* loss-of-function mutants. For statistical details and additional trials see **Supplementary file 1**. (**H**) Schematic of the oxidative stress response pathway in *C. elegans*. Oxidative stress activates SEK-1/MAPKK phosphorylating p38 MAPK phosphorylating the SKN-1/Nrf transcription factor, promoting its nuclear translocation, which initiates transcription of oxidative stress response genes (e.g., *gcs-1* and *gst-4*).

The following source data is available for figure 2:

**Source data 1.** Oxidative stress response genes upregulated by loss of *memo-1* are transcriptional targets of SKN-1.

(**Figure 4B** and **Figure 3—figure supplement 1A**). The level of *hsp-6* expression, a measure of the mitochondrial unfolded protein response and of high mitochondrial ROS levels (**Runkel et al., 2013**), remained at control levels in *memo-1(RNAi)* animals (**Figure 4D**). Moreover, the longevity of *memo-1(RNAi)* treated animals was additive to the longevity of electron transport chain mitochondrial mutant *isp-1(qm150)* (**Supplementary file 1**). Lastly and importantly, a low dose of mitochondrial

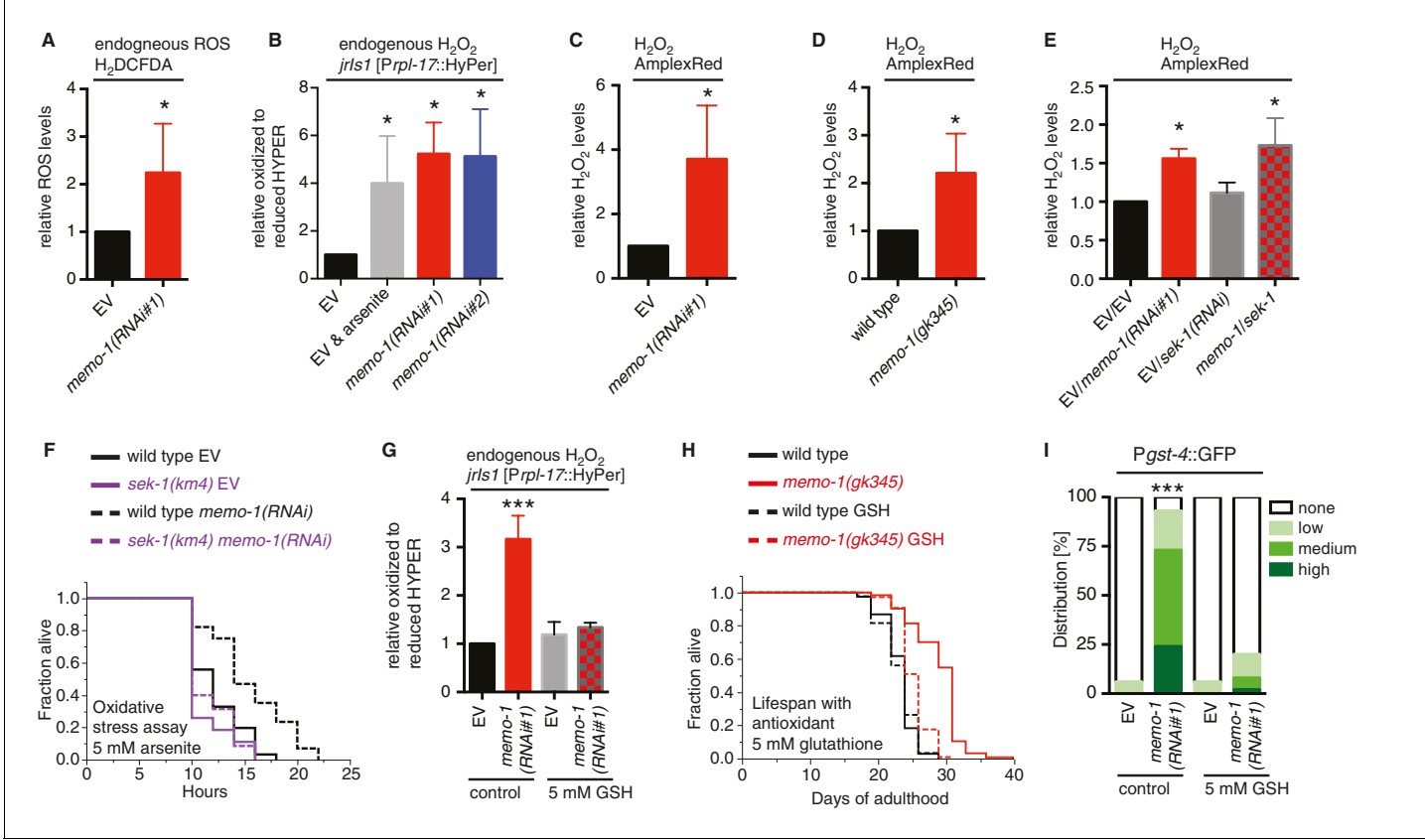

**Figure 3.** Increased ROS resulting from loss of *memo-1* function is required for longevity. (A) Knockdown of *memo-1* increases endogenous ROS levels *in vivo*, measured using the CM-H2DCFDA fluorescent molecular probe. (B) Knockdown of *memo-1* increases endogenous hydrogen peroxide levels, measured in transgenic HyPer worms (*jrIs1*[P*rpl-17*::HyPer]), to a similar extent as measured in EV fed animals treated with 5 mM sodium arsenite for 10 min (EV and arsenite). (C–D) Knockdown of *memo-1* (C) and *memo-1(gk345)* mutants (D) have higher hydrogen peroxide levels *in vivo* compared to wild type (N2) control, as measured with AmplexRed. (E) Knockdown of *sek-1* does not suppress the higher hydrogen peroxide levels induced by *memo-1* *(RNAi)* measured with AmplexRed. (F) Loss of *sek-1* completely suppresses the *memo-1(-)* mediated oxidative stress resistance. For statistical details and additional trials see ***Supplementary file 2***. (G) Treatment with the antioxidant glutathione (GSH) completely suppresses the higher endogenous hydrogen peroxide levels of transgenic HyPer worms (*jrIs1*[P*rpl-17*::HyPer]) treated with *memo-1(RNAi)* or by measurement with Amplex Red (***Figure 3—figure supplement 1C***). Eggs were hatched on empty vector (EV) or *memo-1(RNAi)* food with 5 mM GSH or with solvent (H2O; control) and harvested for assay at larval stage 4 (L4). (H) The antioxidant glutathione (GSH) completely suppresses the longevity of *memo-1(gk345)* mutants. For statistical details and additional trials with *memo-1(RNAi)* see ***Supplementary file 1***. (I) The antioxidant glutathione (GSH) completely suppresses the induction of oxidative stress response gene *gst-4* in response to *memo-1* knockdown. Transgenic animals *dvIs19* [P*gst-4*::GFP] were placed on empty vector (EV) or *memo-1(RNAi)* food with 5 mM GSH or solvent (H2O; control) for two generations and day one adults were scored. N > 60, three merged trials. *** <0.0001 *P* values were determined by Chi$^2$ test. Scoring is described in Material and methods. For (A–C, E) Eggs were hatched on empty vector (EV) or *memo-1(RNAi)* food and harvested for the assay at larval stage 4 (L4). For (A–E, G) N > 1000 for each condition, three merged trials. All data are represented as mean ± s.e.m. *P* values * <0.05 relative to wild-type on EV or (D) to wild type (N2), determined by one sample *t*-test, two-tailed, hypothetical mean of 1.

The following figure supplement is available for figure 3:

**Figure supplement 1.** Reduced *memo-1* function induces NADPH oxidase specific ROS.

ROS require *sod-3* (superoxide dismutase) for extending lifespan (***Yee et al., 2014***) and *memo-1(-)* longevity is independent of *sod-3* (***Supplementary file 1***). Taken together, these data show that the *memo-1(-)* induced ROS are not generated by the mitochondria.

The nicotinamide adenine dinucleotide phosphate oxidases (NADPH oxidase) are important sources of intracellular ROS (***Krause, 2007***). To check if they have a role in ROS production in *memo-1 (RNAi)*-animals, we treated these animals with diphenyleneiodonium (DPI), a general inhibitor of NADPH oxidase activity (***Stuehr et al., 1991***). DPI treatment completely eliminated *memo-1(RNAi)*-

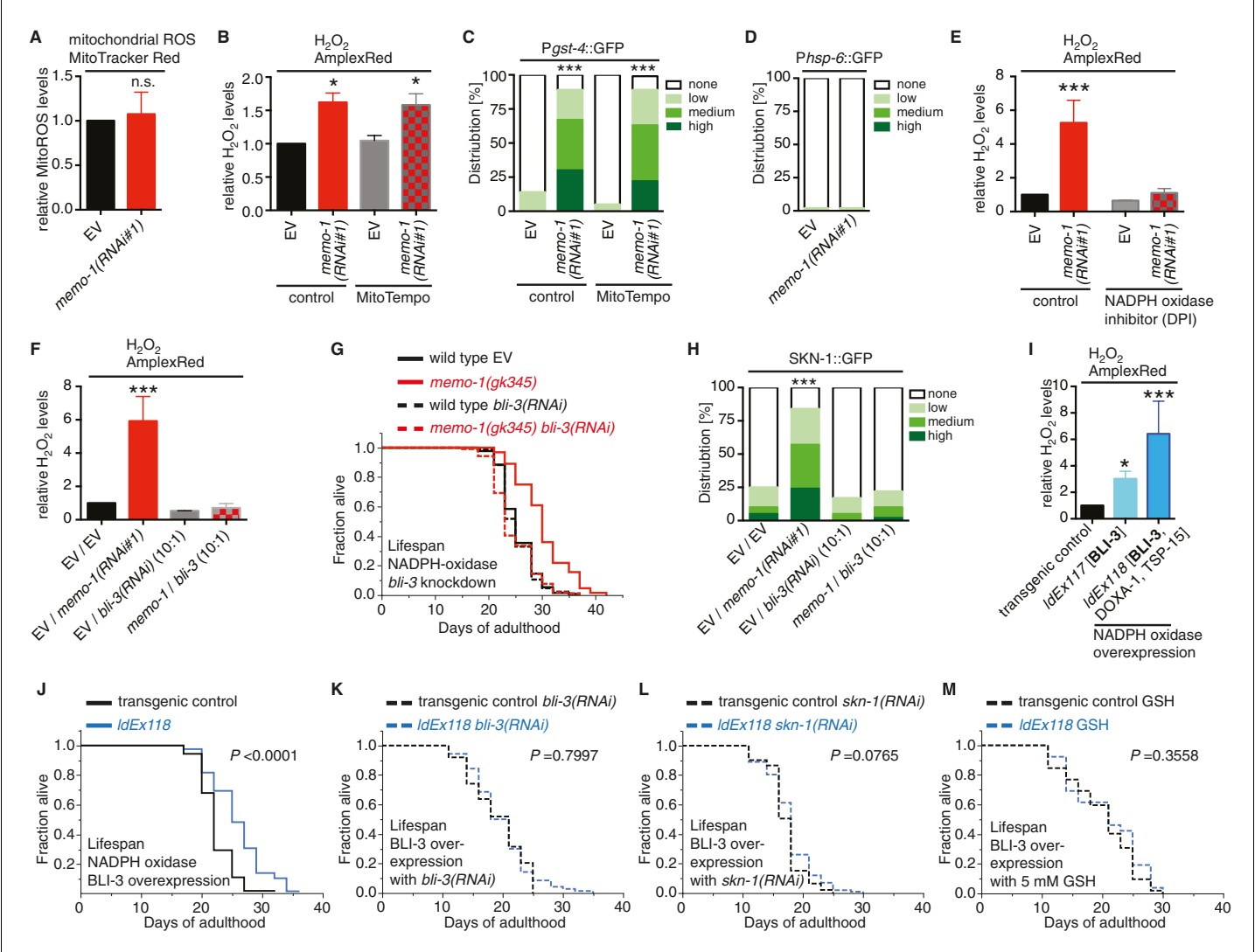

**Figure 4.** Loss of *memo-1* increases ROS via NADPH oxidase to extend lifespan. (**A**) Knockdown of *memo-1* does not increase mitochondrial localized ROS levels *in vivo*, measured with MitoTracker Red CM-H$_2$XRos fluorescent molecular probe. In a parallel experiment, the same *memo-1(RNAi)* treatment was sufficient to increase hydrogen peroxide, as measured by Amplex Red (***Figure 3—figure supplement 1A*** and ***Figure 4—figure supplement 1***). (**B**) The mitochondrial specific antioxidant MitoTempo did not suppress *memo-1(RNAi)* induced ROS, measured with Amplex Red. (**C**) The mitochondrial specific antioxidant MitoTempo did not suppress *memo-1(RNAi))* induced expression of *gst-4* (*dvIs19* [P*gst-4*::GFP]). N > 60, three merged trials. *** <0.0001 *P* values were determined by Chi$^2$ test. Scoring is described in Material and methods. (**D**) The mitochondrial stress response gene *hsp-6* (*zcIs13* [P*hsp-6*::GFP]) was not induced by *memo-1(RNAi)*. N > 100 for each condition, three merged trials. As a control and run in parallel, *memo-1(RNAi)* was sufficient to increase P*gst-4*::GFP expression (***Figure 5—figure supplement 1B***). (**E**) A short (15 min) treatment with the NADPH oxidase inhibitor Diphenyleneiodonium (DPI) completely suppressed the elevated hydrogen peroxide levels of wild-type worms treated with *memo-1 (RNAi)*. (**F**) Knockdown of *bli-3/* NADPH oxidase completely suppressed the elevated hydrogen peroxide levels of wild-type worms treated with *memo-1(RNAi)*. As reported in ***Chávez et al. (2009)***, a 1:10 ratio for the double RNAi treatment with *bli-3* was used because of the strong *bli-3* blistering phenotype. For corresponding experiments with transgenic HyPer worms (*jrIs1*[P*rpl-17*::HyPer]), see ***Figure 3—figure supplement 1D***. (**G**) Adulthood specific knockdown of *bli-3/* NADPH oxidase (undiluted RNAi) completely suppressed the longevity of *memo-1(gk345)* mutants. For statistical details and additional trials see ***Supplementary file 1***. (**H**) Knockdown of *bli-3/* NADPH oxidase completely suppressed *memo-1(RNAi)* induced nuclear translocation of SKN-1 protein (*ldIs007* [SKN-1::GFP]). N > 60, three merged trials. *** <0.0001 *P* values were determined by Chi$^2$ test. Scoring is described in Material and methods. (**I**) Transgenic worms overexpressing BLI-3 (*ldEx117*) alone, or triple transgenic worms (*ldEx118*) overexpressing BLI-3 and co-factors for NADPH oxidase complex maturation and stability (DOXA-1 and TSP-15; [***Moribe et al., 2012***]) showed an increased hydrogen peroxide level compared to control transgenic animals. Mixed stage worms, N > 1000 for each condition, three merged trials. All data are represented as mean ± s.e.m. *P* values * <0.05 or *** <0.0001 relative to transgenic control (*ldEx102*) were determined by one sample *t*-test, two-tailed, hypothetical mean of 1. (**J–M**) Triple transgenic worms (*ldEx118*) overexpressing BLI-3, DOXA-1, TSP-15 showed an increased lifespan compared to transgenic control (*ldEx102*). The increased lifespan was dependent on *bli-3* (**K**), *skn-1* (**L**), and ROS (**M**). (**J–M**) are from the same trial (***Supplementary file 1***). *P*-value determined by log-rank. Overexpression of BLI-3 alone (*ldEx117*) was also sufficient to increase lifespan (***Supplementary file 1***). For statistical

*Figure 4 continued on next page*

*Figure 4 continued*

details and additional trials see *Supplementary file 1*. For (A, B, E, F) eggs were hatched on empty vector (EV) or *memo-1(RNAi)* (or double RNAi with a ratio 1:1, except where indicated) food, and harvested for assay at larval stage 4 (L4). N > 1000 for each condition, three merged trials. All data are represented as mean ± s.e.m. *P* values * <0.05 or *** <0.0001 relative to wild-type on EV were determined by one sample *t*-test, two-tailed, hypothetical mean of 1.

The following figure supplement is available for figure 4:

**Figure supplement 1.** Loss of *memo-1* does not increase mitochondrial ROS.

induced ROS (*Figure 4E*). The *C. elegans* genome encodes two NADPH oxidase (NOX)-related genes, Duox1/BLI-3 and Duox2. NOX proteins contain an NADPH oxidase domain, while DUOX proteins possess an additional peroxidase domain. Duox2 is not expressed in adult *C. elegans*, while BLI-3 is mainly expressed and functional in the intestine and hypodermis (*Edens et al., 2001*; *Chávez et al., 2009*). Importantly, knocking down *bli-3* completely eliminated *memo-1(-)*-induced ROS (*Figure 4F* and *Figure 3—figure supplement 1D*), suggesting that this dual oxidase is required for ROS generation in the *memo-1(-)* animals.

## BLI-3/NADPH oxidase is required and sufficient to extend lifespan via SKN-1

Next we examined the role of BLI-3 in the *memo-1(-)*-induced phenotype. Adult-specific knockdown of *bli-3* did not alter wild-type lifespan or longevity resulting from dietary restriction, reduced insulin/IGF-1 signaling or reduced mitochondrial electron transport chain activity (*Supplementary file 1*), but it completely eliminated the *memo-1(-)* longevity phenotype (*Figure 4G* and *Supplementary file 1*). Consistent with this, *bli-3* knockdown suppressed SKN-1 nuclear localization in *memo-1(RNAi)* animals (*Figure 4H*), suggesting that BLI-3-generated ROS signal to SKN-1. Conversely, overexpression of BLI-3 alone, or with its maturation/association partners DOXA-1 and TSP-15 (*Moribe et al., 2012*), was sufficient to generate higher ROS levels (*Figure 4I*) and to increase lifespan (*Figure 4J* and *Supplementary file 1*), which was reduced by *bli-3 (RNAi)* (*Figure 4K* and *Supplementary file 1*). BLI-3-dependent longevity (*Figure 4J* and *Supplementary file 1*) required SKN-1 (*Figure 4L* and *Supplementary file 1*), and was suppressed by the antioxidant GSH (*Figure 4M* and *Supplementary file 1*). Taken together, these results suggest that MEMO-1 functions as an inhibitor of BLI-3; and elevated ROS in the absence of MEMO-1 confers an enhanced longevity phenotype. This is a novel function for the NADPH oxidase BLI-3, which is known to produce ROS for innate immune functions (*Chávez et al., 2009*).

## MEMO-1 prevents RHO-1/GTPase from activating BLI-3/NADPH oxidase

Loss of *memo-1* increases expression of the SKN-1-target gene *gst-4* in a *bli-3*-dependent manner (*Figure 5A*). By contrast, reduction of insulin/IGF-1 receptor signaling by *daf-2(RNAi)*, which induces *gst-4* via SKN-1 (*Ewald et al., 2015*; *Tullet et al., 2008*), was not blocked by *bli-3(RNAi)* (*Figure 5A* bars 5–7). These findings would suggest that MEMO-1 and DAF-2 act in parallel to each other.

To determine how MEMO-1 inhibits BLI-3 activity, we took a genetic approach. Loss of genes that are required for the downstream signaling in *memo-1(-)* animals should also eliminate *gst-4* upregulation in a manner similar to *bli-3* knockdown (*Figure 5A*). Therefore, we performed a targeted RNAi screen on selected genes. We screened genes that are known to activate NADPH oxidase, as well as genes that either interact with, or might potentially interact with MEMO-1 (*Figure 5—figure supplement 1A–F*). In this screen, we identified known activators of mammalian NADPH oxidases, such as *rac-2*, a Rho family GTPase, *pak-1*, the p21-activated kinase (*Stanley et al., 2014*; *Hurd et al., 2012*), *doxa-1*, the *C. elegans* BLI-3 maturation factor (*Moribe et al., 2012*), *pes-7*/IQGAP, the actin-binding Rho GTPase activator and NRF2 binding partner (*Kim et al., 2013*), and two interactors of mammalian Memo1: the actin-binding depolymerizing factor *unc-60*/cofilin (*Meira et al., 2009*) and the homologue of the mammalian RhoA GTPase *rho-1* (*Stanley et al., 2014*).

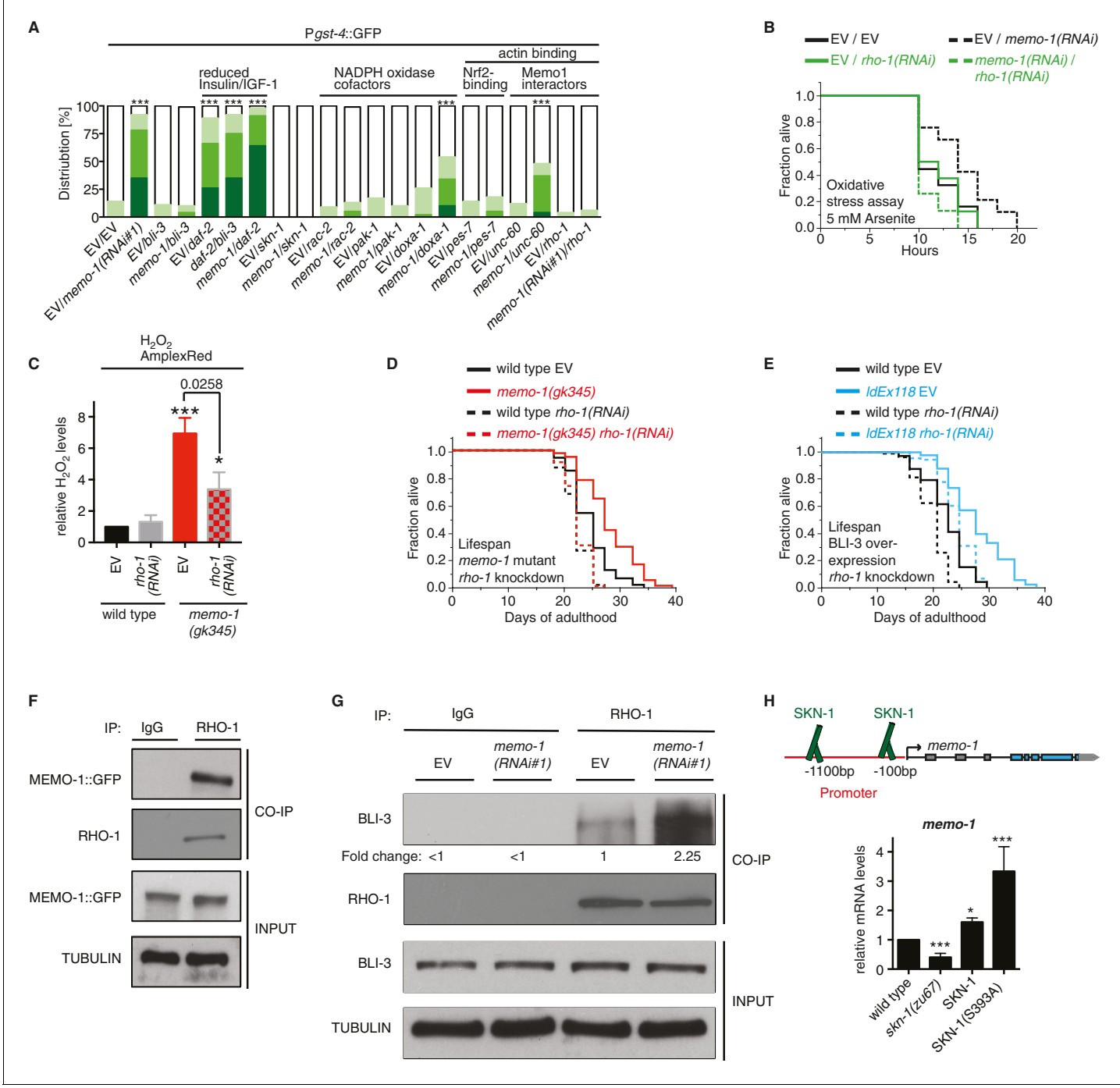

**Figure 5.** MEMO-1 inhibits NADPH oxidase via RHO-1/GTPase. (**A**) Summary of the targeted RNAi screen using the SKN-1 target gene *gst-4* expression (*dvls19* [P*gst-4*::GFP]) as a read-out for *memo-1(-)* induced NADPH oxidase activity. N > 60, three merged trials. *** <0.0001 P values were determined by Chi² test. For individual trials see *Figure 5—figure supplement 1*. Scoring is described in Material and methods. (**B**) Loss of *rho-1* completely suppresses the *memo-1(-)* mediated oxidative stress resistance. For statistical details see *Supplementary file 2*. (**C**) Knockdown of *rho-1* lowered the *memo-1(-)* mediated ROS induction in all three independent biological trials. Eggs were hatched on empty vector (EV) or *rho-1(RNAi)*. All data are represented as mean ± s.e.m. P values * <0.05 and *** <0.0001 relative to wild type EV control were determined by one sample *t*-test, two-tailed, hypothetical mean of 1. The *memo-1(gk345)* mutants treated with *rho-1(RNAi)* was compared to the *memo-1(gk345)* mutants treated with EV using paired *t*-test, one-tailed. (**D**) The longevity of *memo-1(gk345)* mutants was completely suppressed by adulthood specific knockdown of *rho-1*. For statistical details see *Supplementary file 1*. (**E**) The longevity of triple transgenic worms (*ldEx118*) overexpressing BLI-3, DOXA-1, TSP-15 was partially suppressed by adulthood specific knockdown of *rho-1*. For statistical details see *Supplementary file 1*. (**F**) MEMO-1 physically interacts with RHO-1 under wild-type conditions. Lysates from MEMO-1::GFP expressing animals *ldEx112* [MEMOfosmid::GFP] were subjected to immunoprecipitations with

*Figure 5 continued on next page*

*Figure 5 continued*

a *C. elegans* specific RHO-1 antiserum and immunoblotted with a GFP antibody following SDS-PAGE. (**G**) Knockdown of *memo-1* enhances the physical interaction of RHO-1 with BLI-3. Endogenous RHO-1 was immunoprecipitated from lysates of wild type worms fed either EV or *memo-1(RNAi)* using RHO-1 antibody and was immunoblotted with BLI-3 antiserum. The relative BLI-3 levels in Co-IP samples to input samples is expressed as a fold change of *memo-1(RNAi)* to the empty vector control (EV). (**H**) Feedback loop: The *memo-1* promoter contains four predicted SKN-1 binding sites (wormbase. org). In a genome-wide screen, transgenic SKN-1::GFP showed a high signal on two *skn-1*-binding sites in the *memo-1* promoter region (upper panel; [*Niu et al., 2011*]). Lower panel shows that the *memo-1* mRNA levels are lower in *skn-1(zu67)* loss-of-function mutants and higher in conditions of SKN-1 overexpression (*ldIs007* [SKN-1::GFP]) and constitutively nuclear SKN-1 overexpression (*ldIs020* [SKN-1S393::GFP]; [*Tullet et al., 2008*]) in transgenic animals, compared to wild type (N2), determined by qRT-PCR. Three biological samples of each 100 L4 worms per strain per trial. Data are represented as mean ± s.e.m. *P* value * <0.05, ** <0.001, *** <0.0001 relative to wild type or control, by one sample *t*-test, two-tailed, hypothetical mean of 1.

The following figure supplement is available for figure 5:

**Figure supplement 1.** Targeted RNAi screen to discover genes mediating *memo-1* activation of SKN-1.

In mammalian models, it has been shown that RhoA interacts with Memo1 in pull-down assays (*Zaoui et al., 2008*). Moreover, ectopic expression of a gain-of-function mutant RhoA (V14) rescued breast cancer cells from migration-related defects (*Zaoui et al., 2008*). Thus, in the following experiments we concentrated on the MEMO-1/RHOA interaction in worms.

To test whether *C. elegans rho-1* is required for *memo-1(-)*-induced oxidative stress resistance, we knocked-down *rho-1* in *memo-1(RNAi)* animals, which resulted in complete elimination of their oxidative stress resistance (*Figure 5B* and *Supplementary file 2*). Knocking down *rho-1* in *memo-1 (gk345)* mutants partially suppressed the *memo-1(-)* ROS induction (*Figure 5C*). Moreover, adult-specific knockdown of *rho-1* completely suppressed the longevity of *memo-1(gk345),* and partially blocked the lifespan extension from BLI-3 overexpression (*Figure 5D and E* and *Supplementary file 1*). Next, we examined whether MEMO-1 and RHO-1 physically interact in *C. elegans* lysates, by performing co-immunoprecipitation (Co-IP) of endogenous RHO-1 followed by immunoblot analysis of GFP-tagged MEMO-1. Under normal conditions, MEMO-1 and RHO-1 are found in a complex (*Figure 5F*). Since loss of *memo-1* in *C. elegans* leads to higher ROS levels that are dependent on the NADPH oxidase BLI-3, we tested whether RHO-1 and BLI-3 also interact using lysates from control and *memo-1*-impaired animals. Co-immunoprecipitation with a RHO-1 antibody revealed that under normal conditions endogenous RHO-1 physically binds to endogenous BLI-3 *in vivo*. However, we found 2.25-fold higher levels of the BLI-3 bound to RHO-1 in lysates from *memo-1* knockdown worms (*Figure 5G*). These findings suggest that MEMO-1 regulates the RHO-1/BLI-3 interactions necessary for BLI-3 function, a model that is consistent with our genetic findings.

Interestingly, the *memo-1* promoter contains four predicted SKN-1 binding sites (wormbase.org). In a genome-wide screen, transgenic SKN-1::GFP showed a high signal on two of the four sites in the *memo-1* promoter region (*Figure 5H*, upper panel; [*Niu et al., 2011*]), suggesting that SKN-1 regulates *memo-1* transcription. Consistent with this, *memo-1* mRNA levels are lower in *skn-1(zu67)* loss-of-function mutants and, conversely, higher in transgenic animals with either SKN-1 overexpression (*ldIs007* [SKN-1::GFP]), or constitutive nuclear SKN-1 overexpression (*ldIs020* [SKN-1S393::GFP]; [*Tullet et al., 2008*]), when compared to wild type.

These results suggest a model whereby MEMO-1 controls the level of RHO-1 that is bound to BLI-3, and thereby regulates BLI-3 activity (*Figure 6*). In animals with *memo-1* knockdown RHO-1 is free to bind and activate BLI-3/NADPH oxidase, thereby generating an increase in intracellular ROS. These ROS presumably act as a longevity-promoting signal molecule that activates both p38 MAPK and the SKN-1/Nrf1,2,3 transcription factor, which would increase the level of transcripts involved in the oxidative stress response, thus lengthening lifespan. Furthermore, activation of SKN-1 increases the expression of *memo-1* (*Figure 5H*), suggesting a negative feedback loop that regulates ROS generation by the BLI-3/NADPH oxidase (*Figure 6*) in order to maintain a homeostatic balance.

## Discussion

NADPH oxidases are a major source of cellular ROS (*Krause, 2007*), but the possibility that the ROS they produce might influence longevity-associated mechanisms has not been explored. In *C.*

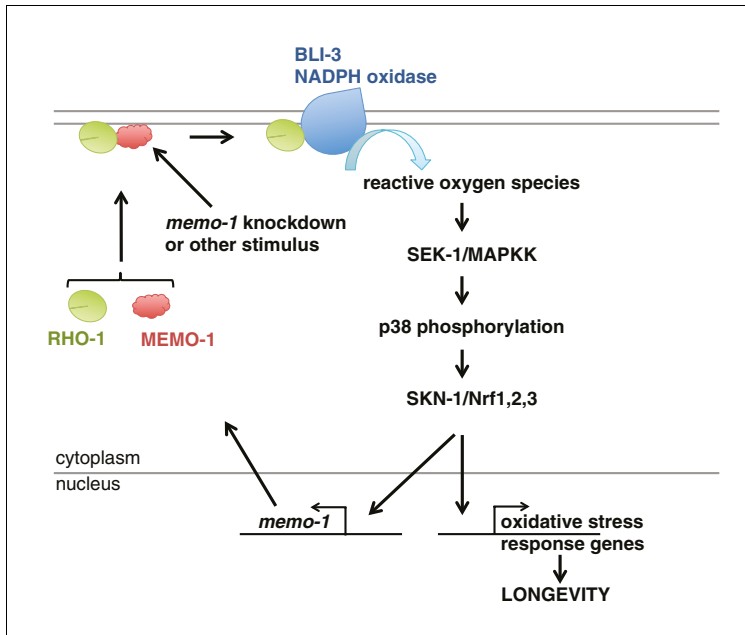

**Figure 6.** Model of how MEMO-1 activates NADPH oxidase activity to promote oxidative stress resistance and longevity. Under normal conditions, MEMO-1 is complexed with RHO-1/GTPase. Loss of *memo-1* frees RHO-1 to enhance BLI-3/NADPH oxidase activity to generate localized ROS, which activates p38 MAPK signaling to SKN-1 to transcribe genes important for oxidative stress resistance and longevity. Similar to ERBB2 recruiting Memo1 in breast cancer cells (*Marone et al., 2004*), we speculate that in *C. elegans* a stimulus might activate a cell-surface receptor to recruit MEMO-1, thereby freeing RHO-1 to promote BLI-3/NADPH oxidase activity. Because SKN-1 also transcribes *memo-1* (*Figure 5H*) resulting in a negative feedback loop to shut off BLI-3/NADPH oxidase activity, *memo-1* RNAi or mutation breaks this feedback loop, resulting in continuously enhanced BLI-3 activity.

*elegans*, we have identified a pathway that regulates ROS production via an increase in BLI-3/ NADPH oxidase activity. We show that MEMO-1 is a negative regulator of BLI-3, and that its loss enhances an interaction between RHO-1 and BLI-3, thereby stimulating the activity of this NADPH oxidase. As a result, ROS are generated that activate p38 and induce nuclear accumulation of SKN-1/Nrf, and thereby promote stress resistance and longevity in this metazoan. Thus, our data identify a regulatory mechanism that controls ROS generation, which is sufficient to promote longevity; they also show that this mechanism initiates an adaptive response similar to that triggered by transient ROS increases generated from the mitochondria.

Our findings support and broaden the emerging hypothesis that directed ROS generation, e.g., in response to specific stimuli or in particular subcellular locations, acts as a homeostatic signal that can have substantial benefits (*Hekimi et al., 2011*; *Shadel, 2014*; *Ristow and Schmeisser, 2014*). The mitochondria generate ROS as a by-product of oxidative phosphorylation, and thereby integrate the rate of metabolism with the animal's physiology. The beneficial effects of low-dose mitochondrial ROS on healthspan and lifespan arise from a hormetic adaptation to mild stress (*Hekimi et al., 2011*; *Shadel, 2014*; *Ristow and Schmeisser, 2014*). Similarly, it was shown recently that ROS production arising from perturbed ER redox homeostasis leads to a protective response that increases lifespan (*Hourihan et al., 2016*). In each of these examples, mutants that lack the capacity to detoxify ROS (e.g. *skn-1(-)* and *sek-1(-)*) exhibit a shortening of lifespan, indicating that these ROS are detrimental in the absence of this response (*Schmeisser et al., 2013b*; *Weimer et al., 2014*; *Schmeisser et al., 2013a*; *Hunt et al., 2011*; *Saul et al., 2010*; *Hourihan et al., 2016*). By contrast, ROS generated by the BLI-3 NADPH oxidase did not further shorten the lifespan of *memo-1* mutant animals that lack the oxidative stress response genes *skn-1* and *sek-1* (*Figure 2A,B and G*). This suggests that the health benefits from NADPH oxidase-generated ROS are initiated by a signaling cascade in the absence of an adaptive stress response to damage, and therefore arise from a ROS signal per se.

A variety of extracellular and intracellular stimuli induce NADPH oxidase activation (*Lambeth and Neish, 2014*). The roles of these enzymes in innate immunity are well-documented, and in *C. elegans* the NADPH oxidase BLI-3 produces ROS in response to pathogen infection, leading in turn to activation of NSY-1/ASK1/MAPKKK, SEK-1/MAPKK, PMK-1/p38 MAPK and SKN-1/Nrf (*Hoeven et al., 2011*). BLI-3 and p38/SKN-1 are also activated by stress from sodium arsenite (*Hourihan et al., 2016*). Importantly, interactions with various cell surface receptors results in NADPH oxidase activation. These include not only receptors involved in immunity signaling but also growth factor receptors (*Lambeth and Neish, 2014*), with the best-described paradigm being the dependence of EGFR signaling on ROS that are produced by NADPH oxidase in response to EGF binding (*Truong and Carroll, 2012*). Our results suggest that all of these signaling events have the potential to impact upon SKN-1/Nrf, and potentially other ROS-responsive mechanisms that influence longevity. Now that we have established MEMO as a regulator of NADPH oxidase, it will be of interest in the future to investigate whether MEMO might be involved in controlling these signaling pathways.

Another exciting subject for future study will be to elucidate how MEMO-1-responsive ROS signals generated by NADPH oxidase act upon the p38 MAPK pathway. Interestingly, NSY-1 and its mammalian homolog ASK-1 are each regulated for activation through cysteine oxidation for activation (*Nadeau et al., 2007*; *Hourihan et al., 2016*), suggesting that this kinase may respond directly to BLI-3-generated ROS. Furthermore, signaling from NADPH oxidases has been shown to activate Nrf2-mediated transcription in mammals (*Sekhar et al., 2003*; *Hecker etal., 2014*; *Brewer et al., 2011*; *Segal et al., 2010*; *Schröder et al., 2012*; *Nlandu Khodo et al., 2012*; *Hourihan et al., 2016*); and RhoA GTPase activity has been linked to NADPH oxidase activation and p38 MAPK signaling (*Kim et al., 2005*). Interestingly, inducing endoplasmic reticulum (ER) stress in HUVEC cultures leads to active RhoA translocation to the ER membrane, resulting in Nox4-dependent ROS generation at this location (*Wu et al., 2010*). Taken together, these findings show a possible conservation of an overall signaling pathway through which mechanisms controlled by MEMO and RhoA activate NADPH oxidase-mediated ROS signaling to act on p38 MAPK and SKN-1/Nrf.

Breast cancer cells that lack Memo1 fail to migrate upon stimulation with growth factors (*Marone et al., 2004*). Moreover, it was shown that Memo1 is required for the localization of GTP-bound RhoA at the plasma membrane (*Zaoui et al., 2008*), an essential step in the process of cell migration. Forcing RhoA to be constitutively membrane-bound rescued the migration phenotype of cells lacking Memo1 (*Zaoui et al., 2008*), suggesting a conserved and biologically relevant interaction between Memo1 and RhoA. We have tried a similar experiment in *C. elegans* to determine whether adult-specific expression of constitutively active RHO-1(G14V) could by-pass *memo-1 function*. We predicted that adult-specific overexpression of RHO-1(G14V) would activate BLI-3/NADPH oxidase leading to an increased lifespan. Unfortunately, overexpression of the constitutively active RHO-1(G14V) GTPase specifically during adulthood causes lethality within 2-4 days (data not shown; [*McMullan and Nurrish, 2011*]). The activity of NADPH oxidase has been shown to be essential during cell migration (*Stanley et al., 2014*; *Hurd et al., 2012*). Interestingly, Memo1 has been linked to NADPH oxidase activity in breast cancer cells (*MacDonald et al., 2014*), however the exact mechanism underlying these results, as well as a role for RhoA, has not been established and is likely to be context-dependent.

Memo1 is a highly conserved protein with homologues found in bacteria through vertebrates (*Schlatter et al., 2012*). The most conserved portion of Memo1 is the domain that has three conserved histidine residues. These three histidines, which are present in the *C. elegans* protein, are required for copper binding, and for Memo1's copper-reducing activity in breast cancer cells (*MacDonald et al., 2014*). One speculative but exciting possibility for a future study would be to investigate whether MEMO-1 might sense ROS through this metal-binding domain, and whether this might provide a feedback mechanism that regulates this ROS-generating pathway.

In the future, it will be important to understand how directed NADPH oxidase activity promotes healthy aging in mammals. In mice, loss of NADPH oxidase activity accelerated aging (*Chen et al., 2015*; *Lee et al., 2011*). By contrast, chronic ROS generation by NADPH oxidase has been suggested to promote age-dependent diseases (*Krause, 2007*), showing that NADPH oxidase and the generated ROS need to be tightly regulated. Our data reveal that transient or low NADPH oxidase-generated ROS act as a signal to re-establish homeostasis during aging, and imply that signals that

affect NADPH oxidase activity can also influence longevity. An exciting aspect of our findings is that NADPH oxidase activity is inducible and controllable, suggesting a novel target for therapeutics to promote healthy aging.

# Materials and methods

## Strains

*Caenorhabditis elegans* strains were maintained on NGM plates and OP50 *Escherichia coli* bacteria at 20°C as described in *Brenner (1974)*. The wild-type strain was N2 Bristol (*Brenner, 1974*). Mutant strains used are described in Wormbase (www.wormbase.org): LGII: *let-23(n1045), rrf-3(pk1426)*; LGIII: *daf-2(e1368)*; LGIV: *skn-1(zu67, zu129)*; LGIV: *isp-1(qm150)*; LGV: *memo-1(gk345)*; LGX: *sek-1 (km4)*. Transgenic strains: CL2166 *dvIs19* [P*gst-4*::GFP; pRF4 *rol-6(su1006gf)*] (*Link and Johnson, 2002*), CF702 *muIs32* [P*mec-7*::GFP; *lin-15(+)*] II (*Ch'ng et al., 2003*), LD001 *ldIs007* [P*skn-1*::SKN-1b/c::GFP; pRF4 *rol-6(su1006gf)*] (*An and Blackwell, 2003*), LD1171 *ldIs003* [P*gcs-1*::GFP; pRF4 *rol-6(su1006gf)*] (*Wang et al., 2010*), LD1252 *ldIs020* [P*skn-1*::SKN-1S393::GFP]; (*Tullet et al., 2008*), *ldEx102* [*rol-6(su1006)*] (*Ewald et al., 2015*), *jrIs1*[P*rpl-17*::HyPer] (*Back et al., 2012b*), SJ4100 *zcIs13* [P*hsp-6*::GFP]V (*Yoneda et al., 2004*), TJ356 *zIs356* [P*daf-16*::DAF-16a/b::GFP; pRF4 *rol-6 (su1006gf)*] (*Henderson and Johnson, 2001*), QZ16 *muIs32* [P*mec-7*::GFP; *lin-15(+)*] II; *memo-1 (gk345)* V, QZ25 *muIs32* [P*mec-7*::GFP; *lin-15(+)*] II; *eri-1(mg366)* IV; *lin-15b(n744)* X, QZ50 *jxEx8* [P*memo-1*::GFP; Pmyo-3::RFP].

## *memo-1(gk345)* knock out mutant

*memo-1(gk345)* mutants were obtained from *Caenorhabditis* Genetics Center (CGC; Strain VC794) and were outcrossed eight times against wild type (N2). The *gk345* deletion in *memo-1* deletes the transcription initiation ATG from isoform A (C37C3.8a), but not isoform B (C37C3.8b). The *gk345* deletion causes a frame shift leading to an early stop codon in isoform B (C37C3.8b), which might express a truncated peptide of 30 amino acids. However, only the first 18 out of these 30 amino acids are shared with the wild-type MEMO-1 protein. Furthermore, the *gk345* deletion covers 2 out the five important amino acids of the copper-binding/enzymatic binding site and therefore any possible splice variant might not be functional. This suggests that *gk345* deletion could potentially be a null allele of *memo-1*.

## Construction of transgenic lines

Construction of transcriptional fusion of *memo-1* promoter with GFP (*jxEx8* [P*memo-1*::GFP]). The cis regulatory regions of *memo-1*/C37C3.8 were amplified by PCR from cosmid C37C3 (Accession number U64857) using memo-GFP fusion primers as described in *Hobert (2002)*. PCR for 5′ cis region of *memo-1* (PCR1): Forward primer (memo-GFP-5for) and reverse primer (memo-GFP-5rev-int) that specifically recognizes the 5′ end of the C37C3.8 locus are used to amplify the 5′ cis region of the *memo-1* gene and the first 48 amino acids of the b isoform of the gene (3.5 kb). PCR for 3′ cis region of *memo-1* (PCR2): Forward primer (memo-GFP-3for:) and a reverse primer (memo-GFP-3rev-int) that specifically recognizes the 3′ end of the C37C3.8 locus are used to amplify the 3′ untranslated region (UTR) and the 3′ cis region of the *memo-1* gene (264 bp). PCR3: The green fluorescent protein (GFP) open reading frame (1.4 kb) was amplified from L3691 plasmid (pPD117.01 Fire Lab Vector Kit Documentation 1997. (www.addgene.org)) using fusion primers (gfp-memo-for and gfp-memo-rev). The PCR product from PCR3 was then fused to the 5′ cis memo PCR product from PCR1, as well as to the 3′ cis memo PCR product from PCR2 in a third round of PCR reactions. The resulting two fragments from the third round of PCR reactions are then fused into one final PCR product (P*memo-1*::GFP) in a fourth round of reaction. Primer sequences are found in *Supplementary file 3*). This fused PCR fragment of P*memo-1*::GFP (50 ng/µl together with 100 ng/µl of P*myo-3*::RFP) was injected into wild-type (N2) worms and three transgenic [P*memo-1*::GFP] lines were obtained, of which one was chosen for analysis (QZ50 *jxEx8* [P*memo-1*::GFP; Pmyo-3::RFP]).

## Generation of transgenic lines

Overexpression of MEMO::GFP: 35 ng/µl *tag-253* fosmid (WRM0639B_H07(pRedFlp-Hgr)(tag-253 [30783]::S0001_pR6K_Amp_2xTY1ce_EGFP_FRT_rpsl_neo_FRT_3xFlag)dFRT::unc-119-Nat         from

https://transgeneome.mpi-cbg.de/transgeneomics/index.html) together with 100 ng/µl pRF4 *rol-6* *(su1006gf)* was injected into wild-type (N2) worms and four transgenic *ldEx112-6* [MEMOfosmid:: GFP] lines were obtained, of which one was chosen for picture analysis and CoIP (*ldEx112* [MEMO-fosmid::GFP; *rol-6(su1006gf)*]).

Overexpression of BLI-3: 5 ng/µl of plasmid pHM363 [BLI-3] (*Moribe et al., 2012*) together with 100 ng/µl pRF4 *rol-6(su1006gf)* was injected into wild-type (N2) worms and one transgenic *ldEx117* [BLI-3] line was obtained for analysis.

Triple transgenic overexpression (BLI-3, DOXA-1, TSP-15): 5 ng/µl of plasmid pHM363 [BLI-3] (*Moribe et al., 2012*), 5 ng/µl of plasmid pHM327 [DOXA-1::Venus] (*Moribe et al., 2012*), 5 ng/µl of plasmid pHM106 [HisXp::TSP-15] (*Moribe et al., 2012*), together with 100 ng/µl pRF4 *rol-6* *(su1006gf)* was injected into wild-type (N2) worms and one transgenic *ldEx118* [BLI-3, DOXA-1, TSP-15] lines were obtained for analysis.

## Knockdown by RNA interference

RNAi#1 clone is from Vidal RNAi library (*Rual et al., 2004*) and RNAi#2 clone is from Ahringer RNAi library (*Fraser et al., 2000*; *Kamath et al., 2003*). RNAi bacteria cultures were grown overnight in LB with carbenicillin [100 µg/ml] and tetracycline [12.5 µg/ml], diluted to an OD600 of 1, and induced with 1 mM IPTG and spread onto NGM plates containing tetracycline [12.5 µg/ml] and ampicillin [50 µg/ml]. For empty RNAi vector (EV) plasmid pL4440 was used as control. For double RNAi: bacterial cultures were grown separately and then mixed in a 1:1 ratio, except for combinations with *bli-3* *(RNAi)*. As reported in *Chávez et al. (2009)*, a 1:10 ratio for *bli-3* to other bacteria clone for the double RNAi treatment was used because of the strong *bli-3* blistering phenotype.

## Measuring total ROS levels with fluorescent probe

The total level of ROS were measured by using either a small diffusible fluorescent probe (chloromethyl derivative; CM-$H_2$DCFDA; Life Technologies) or with the Amplex Red Hydrogen Peroxide/ Peroxidase Assay Kit (Life Technologies) with a protocol adapted from *Schulz et al. (2007)*. Briefly, wild-type eggs were hatched on empty vector (EV) or *memo-1(RNAi)* food and harvested for assay at larval stage 4 (L4). More than 1000 L4 animals per conditions were harvested into 96-well plates, incubated in 50 µM CM-$H_2$DCFDA for 1 hr and the fluorescent intensity was measured with an excitation wavelength of 485 nm and a 520 nm emission filter. Detection of ROS by Amplex Red was performed according to manufacturer's instructions and fluorescence was measured with an excitation wavelength of 530 nm and a 590 nm emission filter. Animals were lysed and protein levels were determined using BCA assay (Pierce). The fluorescent intensity was normalized to protein levels and is indicated relative to control.

## Measuring endogenous hydrogen peroxide levels with transgenic HyPer worms

To measure endogenous hydrogen peroxide levels, transgenic *jrIs1*[P*rpl-17*::HyPer] worms expressing a HYPER-probe were used as described in *Back et al. (2012b)*. Transgenic *jrIs1*[P*rpl-17*::HyPer] animals were treated with RNAi and about 1000 L4 worms were harvested in 96 microtiter well plates. An excitation wavelength of either 490 nm or 405 nm was used to measure oxidized or reduced HyPer probe fluorescence respectively with an emission filter of 535 nm. The animals were harvested and protein levels were determined using BCA assay (Pierce). The fluorescence intensity was normalized to protein and the ratio of oxidized/reduced HyPer levels was calculated using the 405 nm fluorescence as a numerator and the 490 nm fluorescence as the denominator as described in *Back et al. (2012b)*.

## Antioxidant treatment

For GSH treatment, NGM plates containing 5 mM GSH (Sigma) were poured and allowed to dry. The respective *E.coli* food source was added to each plate and once dry, animals (young adult) were added to the plate and allowed to lay eggs. Nematodes grew for one generation on these plates prior to ROS analysis or GFP scoring. Inhibition of BLI-3 by DPI (Diphenyleneiodonium chloride, Sigma) was performed by treating animals with 80 µM of the compound in M9 solution for 15 min prior to analysis. Removal of mitoROS by the mitochondrial-targeted superoxide scavenger Mito-

Tempo (Sigma) occurred by exposing animals to 250 µM of the antioxidant for 2 hr in M9 solution prior to ROS determination.

## Measuring mitoROS

Wild-type eggs were hatched on the respective *E. coli* food source and harvested for mitoROS analysis at larval stage 4 (L4). As a positive control, wild-type animals were treated with 3 mM Anitmycin A for 2 hr, by top-coating the culturing plates containing the L4 and bacterial food. Approximately 1000 L4 animals per condition were washed and added into each well of a 96-well plate, incubated in 50 µM Mitotracker Red CM-H$_2$XRos (Life Technologies) for 2 hr in the dark on a shaker and the fluorescent intensity was measured with an excitation wavelength of 570 nm and a 610 nm emission filter. Subsequently, animals were lysed and the total protein levels were determined using the BCA assay (Pierce). The level of fluorescence was normalized to protein levels and is expressed relative to control.

## Lifespan assays

Adult lifespan was determined either with or without 5-Fluoro-2'deoxyuridine (FUdR), a DNA synthesis inhibitor that blocks progeny production and thus prevents overcrowding. For non-FUdR lifespan assays: About 100 L4 worms per strain were picked on NGM plates containing OP50 bacteria. Worms were transferred every day or every other day on fresh NGM OP50 plates until progeny production ceased as described in *Kenyon et al. (1993)*, *Alcedo and Kenyon (2004)*. For FUdR lifespan assays: About 100 L4 worms per strain were picked on NGM plates containing OP50 bacteria. The next day, worms (day-1-adults) were transferred onto either NGM plates containing 400 µM FUdR and OP50 bacteria or RNAi bacteria (or plates that also contained 5 mM GSH, or OP50 that were heat-killed by 1 hr of incubation at 75°C before seeding). Animals were scored every day or every other day. All lifespans were plotted with L4 as time-point = 0. Animals were scored as dead animals if they failed to respond to prodding. Exploded or bagged animals and animals that ran off the plate were excluded from the statistics. The estimates of survival functions were calculated using the product-limit (Kaplan-Meier) method. The log-rank (Mantel-Cox) method was used to test the null hypothesis and calculate *P* values (JMP software v.9.0.2.).

## Oxidative stress assays

For arsenite stress tolerance assay: about 50 L4 worms per strain/ condition were picked onto fresh OP50 plates. The next day, 10–12 day-one worms were picked into 24 well plates containing 50 µl physiological M9 Buffer in quadruplicates for each strain and condition (four wells). Then when all worms were set up, 1 mL M9 buffer with either a final concentration of 5 mM or 10 mM sodium arsenite (Sigma-Aldrich) was added to three wells per strain and condition. In the remaining well only 1 mL M9 buffer was added as a control. The survival was scored every hour until all died. Exploded animals were excluded from the statistics. For *tert*-Butyl hydroperoxide (*t*-BOOH) stress tolerance assay: about 80 L4 worms per strain/ condition were picked onto fresh OP50 plates. Three days later, 20 day-three worms were picked onto NGM plates containing 15.4 mM *t*-BOOH (Sigma-Aldrich). Because worms try to avoid the *t*-BOOH, they leave the plate. For the first two hours worms were continually repositioned into the middle of the plate until they ceased crawling. Survival was scored every hour. Exploded animals or animals damaged from moving were censored from the statistics. For both arsenite and *t*-BOOH the estimates of survival functions were calculated using the product-limit (Kaplan-Meier) method. The log-rank (Mantel-Cox) method was used to test the null hypothesis and calculate *P* values (JMP software v.9.0.2.). Both oxidative stress tolerance assays are described in more detail at Bio-protocol (*Ewald et al., 2017*).

## Quantitative real-time polymerase chain reaction (qRT-PCR) assays

RNA was isolated with Trizol (TRI REAGENT Sigma), DNAse-treated, and cleaned over a column (RNA Clean and Concentrator ZYMO Research). First-strand cDNA was synthesized in duplicate from each sample (Invitrogen SuperScript III). SYBR green was used to perform qRT-PCR (ABI 7900). For each primer set, a standard curve from genomic DNA accompanied the duplicate cDNA samples (*Glover-Cutter et al., 2008*). mRNA levels relative to N2 control were determined by normalizing to the number of worms and the geometric mean of three reference genes (*cdc-42*, *pmp-3*, and

Y45F10D.4; [Hoogewijs et al., 2008]). At least two biological replicates were examined for each sample. For statistical analysis, one sample t-test, two-tailed, hypothetical mean of 1 was used for comparison using Prism 6.0 software (GraphPad).

## Scoring of transgenic promoter driven GFP or SKN-1::GFP nuclear localization

For each assay, two investigators scored transgenic animals blindly after mounting on slides or with stereomicroscope essentially as described in Robida-Stubbs et al. (2012). Pgcs-1::GFP (Figure 1I) and Pgst-4::GFP (Figure 1G–H, 3I, 4C and 5A, Figure 5—figure supplement 1): At the L4 stage, ldIs003 [Pgcs-1::GFP] transgenic animals were mounted on slides and GFP fluorescence in the intestine was scored using Zeiss AxioSKOP2 microscope at 40x as described in Wang et al. (2010). None: no GFP in intestine. Low: only anterior or posterior intestine with GFP. Medium: both anterior and posterior intestine with GFP but no GFP in the middle of the intestine. High: GFP throughout the intestine. P values were determined by Chi$^2$ test.

Pgst-4::GFP: At the first day of adulthood stage, transgenic dvIs19 [Pgst-4::GFP] (Link and Johnson, 2002) animals were scored using stereoscope described in Wang et al. (2010). None: no GFP in intestine. Low: only weak anterior or posterior intestine with GFP. Medium: both anterior and posterior intestine with strong GFP but no or weak GFP in the middle of the intestine. High: GFP throughout the intestine. P values were determined by Chi$^2$ test. SKN-1::GFP nuclearlocalisation assay (Figure 2D and 4H): Transgenic ldIs007 [Pskn-1::SKN-1b/c::GFP] (An and Blackwell, 2003) L4 animals were mounted on slides and intestinal GFP scored as described in Robida-Stubbs et al. (2012) by using a Zeiss AxioSKOP2 microscope. none = no GFP observed in intestinal nuclei; low = some intestinal nuclei show GFP; medium = more than half of the intestinal nuclei show GFP; high = all intestinal nuclei show GFP. SKN-1::GFP is constitutively expressed in ASI neurons (An and Blackwell, 2003) and we did not observe any noticeable change of SKN-1::GFP expression in ASI upon memo-1 knockdown. P values were determined by Chi$^2$ test using Prism 6.0 software (GraphPad).

## Co-immunoprecipitation and immunoblotting

Approximately 5000 mixed-stage nematodes were sonicated in IP buffer (50 mM Tris-HCl (pH 7.5), 150 mM NaCl, 1% NP-40, 2 mM EDTA and protease inhibitor) and kept on ice for 30 min before being centrifuged for 20 min at 13,000x g. The supernatant was pre-cleared with 50 µl slurry of Protein-G beads for 30 min at 4°C, centrifuged for 5 min at 2000 x g and the beads were removed. The supernatant (1 mg) was incubated with 2 µg of antibody overnight at 4°C. The following day, 50 µl slurry of Protein G beads was added to the lysate for 2 hr at 4°C after which the beads were washed three times in IP buffer and eluted in 2 x sample buffer (+2ME). SDS-page was performed with 10% Bis-Tris gels (Invitrogen) and proteins were transferred to nitrocellulose membranes (Pierce). Western blot analysis was performed under standard conditions with antibodies against GFP, RHO-1 (gift from A. Sugimoto), Tubulin (Cell signal Tech.), BLI-3 (gift from E. Mekada), and Phospho-p38 (Cell signal Tech.). HRP-conjugated rabbit anti-mouse and goat anti-rabbit secondary antibodies were purchased from Sigma and proteins were detected by enhanced chemiluminescence (Pierce). Quantification of protein levels was determined using ImageJ software and normalized to loading control.

## Acknowledgements

We thank Bart P. Braeckman for jrIs1 strain, Hiroki Moribe for bli-3, doxa-1, and tsp-15 plasmids and BLI-3 antibody, Asako Sugimoto for RHO-1 antibody, Alcedo, Blackwell, and Hynes lab members for helpful discussions. VC794 strain was provided by the CGC, which is funded by the NIH office of Research Infrastructure Programs (P40 OD010440). Supported by funding from the Novartis Research Foundation and the Swiss National Science Foundation 310030 A-121574 and 310030B-138674 to NEH and PBSKP3_140135, P300P3_154633, and PP00P3_163898 to CYE and NIH GM062891 and GM094398 to TKB.

# Additional information

## Funding

| Funder | Grant reference number | Author |
|---|---|---|
| Schweizerischer Nationalfonds zur Förderung der Wissenschaftlichen Forschung | PBSKP3_140135 | Collin Yvès Ewald |
| Novartis Stiftung für Medizinisch-Biologische Forschung | | Nancy E Hynes |
| National Institutes of Health | GM062891 | T Keith Blackwell |
| Schweizerischer Nationalfonds zur Förderung der Wissenschaftlichen Forschung | P300P3_154633 | Collin Yvès Ewald |
| Schweizerischer Nationalfonds zur Förderung der Wissenschaftlichen Forschung | 310030A-121574 | Nancy E Hynes |
| Schweizerischer Nationalfonds zur Förderung der Wissenschaftlichen Forschung | 310030B-138674 | Nancy E Hynes |
| National Institutes of Health | GM094398 | T Keith Blackwell |
| Schweizerischer Nationalfonds zur Förderung der Wissenschaftlichen Forschung | PP00P3_163898 | Collin Yvès Ewald |

The funders had no role in study design, data collection and interpretation, or the decision to submit the work for publication.

## Author contributions

CYE, Performed almost all experiments, except western blots, Co-IPs, or dauer assays, Other authors contributed biological repeats of most experiments, Generated all new transgenic lines, Wrote the manuscript in consultation with the other authors, Conception and design, Acquisition of data, Analysis and interpretation of data, Drafting or revising the article; JMH, Performed oxidative stress assays, Scored GFP reporters, Did all western blots and Co-IPs, Conception and design, Acquisition of data, Analysis and interpretation of data, Drafting or revising the article; MSB, Performed lifespan assays, Conception and design, Acquisition of data, Analysis and interpretation of data, Drafting or revising the article; CO, Performed lifespan and oxidative stress assays and scored GFP reporters, Conception and design, Acquisition of data, Analysis and interpretation of data, Drafting or revising the article; IK, Performed dauer assays., Conception and design, Acquisition of data, Analysis and interpretation of data, Drafting or revising the article; LEMM, Performed lifespan assays, Acquisition of data, Analysis and interpretation of data; JA, Performed oxidative stress assays, Conception and design, Acquisition of data, Analysis and interpretation of data, Drafting or revising the article, Contributed unpublished essential data or reagents; TKB, Conception and design, Analysis and interpretation of data, Drafting or revising the article, Contributed unpublished essential data or reagents; NEH, Wrote the manuscript in consultation with the other authors, Conception and design, Analysis and interpretation of data, Drafting or revising the article, Contributed unpublished essential data or reagents

## Author ORCIDs

Collin Yvès Ewald, http://orcid.org/0000-0003-1166-4171

# Additional files

## Supplementary files

• Supplementary file 1. Loss of memo-1 increases adult lifespan.

• Supplementary file 2. Loss of memo-1 increases oxidative stress resistance.

• Supplementary file 3. Primer Sequences.

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
