## [Decision Letter]

Thank you for submitting your article "NADPH-oxidase-mediated redox signaling promotes oxidative stress resistance and longevity through Memo" for consideration by *eLife*. Your article has been favorably evaluated by Tony Hunter (Senior Editor) and three reviewers, one of whom, Andrew Dillin (Reviewer #1), is a member of our Board of Reviewing Editors.

The reviewers have discussed the reviews with one another and the Reviewing Editor has drafted this decision to help you prepare a revised submission.

Summary of work presented:

In this body of work the authors begin to explore how an interactor in the ErbB-2 tyrosine kinase pathway might contribute to aging in the nematode *C. elegans*. The closest ortholog of the mediator of ErbB2-driven cell motility factor, *memo-1*, results in increased lifespan when knocked down or mutated. This appears independent of dietary intake, insulin/IGF1 pathway or germline mediated responses. The interaction with the mitochondrial ETC pathway is sparsely explored and detailed more below.

Transcript analysis indicates that several antioxidant genes are induced by *memo-1* RNAi and lead to the hypothesis that *skn-1* might play a role in the *memo-1* mediated lifespan response. Genetic interaction studies and nuclear localization of *skn-1* support this hypothesis. Furthermore, ROS levels using 3 different sensors all indicate that ROS levels are increased. GSH treatment also blocks *memo-1* (RNAi) lifespan. However, the authors then go on to interrogate whether the ROS levels originate from mitochondria using mitotracker red and MitoTempo, both of which provide negative results without positive controls.

With a lack of correlation of ROS from mitochondria, the authors speculate that ROS is generated from another source and hypothesize that NAPDPH oxidase could be the culprit. Genetic studies of *bli-3* (NADPH oxidase) RNAi confirms their hypothesis in test of ROS levels and longevity. Importantly, *bli-3* overexpression increases ROS and increases lifespan.

Understanding how *bli-3* is regulated by *memo-1*, the authors query orthologes of known mammalian interactors of *memo-1* in a screen and uncover that loss of rho mirrors the effects seen with *bli-3* (although overexpression studies of rho are not performed).

While all reviewers remain enthusiastic about this body of work, a summary of additional experiments to strengthen the claims and interpretation of the results is provided:

1) Because of the role of this pathway in innate immunity, the authors need to explore if the lifespan effects are due to bacterial infection/interaction. It is suggested to perform the lifespan extending experiments on dead bacteria (*memo-1* loss and *bli-3* overexertion).

2) The involvement of mitochondrial produced ROS is not well controlled and we suggest that positive controls in the measurements are needed. *isp-1* mutant animals are documented to have increased ROS and are a good place to start.

3) The involvement of the ETC longevity pathway is not explored and it is suggested to test *bli-3* RNAI on cco-1(RNAi) treated animals or *isp-1* mutant animals for lifespan extension. The same is true for *memo-1* loss.

4) The closing experiments involving rho are not fully executed and do not fully test the model proposed. Does loss of rho block the ROS accumulation in *memo-1*(-) animals? Does loss of rho specifically reduce the lifespan of *memo-1*(-) or *bli-3*(o/e) animals? Does overexertion of rho extend lifespan, and is this *bli-3* dependent?

Finally, the title needs to include *C. elegans*.

---

## [Author Response]

*[…] While all reviewers remain enthusiastic about this body of work, a summary of additional experiments to strengthen the claims and interpretation of the results is provided:*

*1) Because of the role of this pathway in innate immunity, the authors need to explore if the lifespan effects are due to bacterial infection/interaction. It is suggested to perform the lifespan extending experiments on dead bacteria (memo-1 loss and bli-3 overexertion).*

We thank the reviewer for this suggestion. We investigated whether *memo-1* mutants live longer when fed heat-killed OP50, and included the findings in the text:

“Since *sek-1* and *skn-1* are important for the defense against pathogenic bacteria (Hoeven et al. 2011), and bacterial proliferation in the *C. elegans’* intestine contributes to death of the animal (Garigan et al. 2002), we investigated the lifespan of *memo-1* mutants on heat-killed bacteria. We found that *memo-1* mutants remain long-lived compared to wild type ([Supplementary-material SD2-data]).”

*2) The involvement of mitochondrial produced ROS is not well controlled and we suggest that positive controls in the measurements are needed. isp-1 mutant animals are documented to have increased ROS and are a good place to start.*

We thank the reviewer for pointing this out. We have repeated mitochondrial ROS assays and as a positive control used Antimycin A, which binds cytochrome c reductase and thereby disrupts the proton gradient across the inner mitochondrial membrane, which results in the formation of mitochondrial ROS. For 2 of the 5 biological independent trials, we split our animal populations in half and used the corresponding half to measure hydrogen peroxide in the supernatant with Amplex Red, in parallel to the mitochondrial ROS measurements with MitoTracker Red CMXRos, as we did in the previous experiments. Antimycin A increased ROS levels, as measured with both MitoTracker Red and Amplex Red. Mutants that lack *memo-1* showed an almost 2-fold increase in ROS measured with Amplex Red, but showed no significant difference in mitochondrial ROS compared to wild type measured with MitoTracker Red. This argues that the mitochondria are not the source of ROS in animals that lack *memo-1*. We included these data in Figure 4—figure supplement 1.

*3) The involvement of the ETC longevity pathway is not explored and it is suggested to test bli-3 RNAI on cco-1(RNAi) treated animals or isp-1 mutant animals for lifespan extension. The same is true for memo-1 loss.*

As suggested, we have performed the lifespan of *isp-1* mutants treated with *bli-3* RNAi and *memo-1* RNAi. The *bli-3(RNAi)* did not suppress and *memo-1(RNAi)* was additive to the longevity of *isp-1* mutants, suggesting an independent or parallel pathway to the ETC longevity pathway ([Supplementary-material SD2-data]). We included this in the text:

“Moreover, the longevity of *memo-1(RNAi)* treated animals was additive to the longevity of electron transport chain mitochondrial mutant *isp-1(qm150)* ([Supplementary-material SD2-data]).”

“Next we examined the role of BLI-3 in the *memo-1(-)*-induced phenotype. Adult-specific knockdown of *bli-3* did not alter wild-type lifespan or longevity resulting from dietary restriction, reduced insulin/IGF-1 signaling or reduced mitochondrial electron transport chain activity ([Supplementary-material SD2-data]), but completely eliminated the *memo-1(-)* longevity phenotype (Figure 4 and [Supplementary-material SD2-data]).”

*4) The closing experiments involving rho are not fully executed and do not fully test the model proposed. Does loss of rho block the ROS accumulation in memo-1(-) animals?*

Knockdown of *rho-1* in *memo-1(-)* mutants partially suppressed the ROS accumulation in *memo-1(-)* mutants (Figure 5). The partial suppression might be because RNAi does not cause a complete loss of expression, and because wild type animals fed *rho-1* RNAi exhibited higher ROS levels in 2 out of 3 trials. This suggests that *rho-1* knockdown may, through an unknown mechanism, increase ROS on its own.

*Does loss of rho specifically reduce the lifespan of memo-1(-) or bli-3(o/e) animals?*

In two independent trials, adulthood *rho-1(RNAi)* completely suppressed *memo-1(-)* mutant longevity (Figure 5, [Supplementary-material SD2-data])Furthermore, adulthood *rho-1(RNAi)* partially suppressed longevity of BLI-3 overexpressing animals (1 independent trial; Figure 5, [Supplementary-material SD2-data]). This indicates that *rho-1* makes a major contribution to *bli-3*-mediated longevity, and supports our models. The partial dependence could be due to the fact that *rho-1* function is unlikely to be completely eliminated by *rho-1* RNAi treatment during adulthood. RHO-1 has many important functions, and null mutants of *rho-1* are embryonic lethal. To bypass *rho-1* requirements during development, Rachel McMullan and Stephen Nurrish have generated transgenic animals that can drive an inhibitor of endogenous Rho (C3 transferase) by an inducible heat-shock promotor. Unfortunately, adult-specific expression this Rho inhibitor caused lethality within 2-3 days (57). Therefore, we were not able to investigate a complete loss of *rho-1* function during adulthood.

We incorporated these data and added the following sentences in the manuscript.

“Knocking down *rho-1* in *memo-1(gk345)* mutants partially suppressed the *memo-1(-)* ROS induction (Figure 5). Moreover, adult-specific knockdown of *rho-1* completely suppressed the longevity of *memo-1(gk345),* and partially blocked the lifespan extension from BLI-3 overexpression (Figure 5 and [Supplementary-material SD2-data]).”

*Does overexertion of rho extend lifespan, and is this bli-3 dependent?*

Overexpression of the constitutively active RHO-1(G14V) GTPase specifically during adulthood causes lethality within 2-4 days (McMullan & Nurrish 2011). Dr. Stephen Nurrish kindly gave us these transgenic animals (QT54 *nzIs1* [P*hsp-16.2*::RHO-1(G14V)]), in which RHO-1(G14V) expression is driven by a heat-shock promoter. We tried very hard to establish conditions whereby weaker activation of the heat-shock promoter (e.g. at 25^o^C) would lead to lower levels of RHO-1(G14V) expression that might not be toxic, but even under these conditions RHO-1(G14V) overexpression during adulthood shortened lifespan (not shown). This result is not surprising given the many functions of RHO-1 in cellular architecture and signaling. We hope that the reviewer will agree that this result does not contradict our models, particularly in light of our other data supporting the involvement of RHO-1 in BLI-3-mediated longevity.

*Finally, the title needs to include C. elegans.*

*C. elegans* is now included in the title.